# Fast and Accurate Fair $k$-Center Clustering in Doubling Metrics

## ABSTRACT

We study the classic $k$-center clustering problem under the additional constraint that each cluster should be *fair*. In this setting, each point is marked with one or more *colors*, which can be used to model protected attributes (e.g., gender or ethnicity). A cluster is deemed *fair* if, for every color, the fraction of its points marked with that color is within some prespecified range. We present a coreset-based approach to fair $k$-center clustering for general metric spaces which attains almost the best approximation quality of the current state of the art solutions, while featuring running times which can be orders of magnitude faster for large datasets of low doubling dimension. We devise sequential, streaming and MapReduce implementations of our approach and conduct a thourough experimental analysis to provide evidence of their practicality, scalability, and effectiveness.

**ACM Reference Format:**
Anonymous Author(s). 2023. Fast and Accurate Fair $k$-Center Clustering in Doubling Metrics. In *Proceedings of ACM Conference (Conference'17)*. ACM, New York, NY, USA, 11 pages. https://doi.org/10.1145/nnnnnnn.nnnnnnn

## 1 INTRODUCTION

Clustering, in its many variants, is a fundamental primitive in unsupervised learning and data analysis, aiming at grouping points according to some notion of similarity. In the most common setting, the input to clustering is a set of points $S$ from a metric space $(M, d)$, where $d : M \times M \to \mathbb{R}_0^+$ is a distance function, modeling dissimilarity [20]. A popular variant is $k$-clustering, which requires to select a set of $k$ centers and to build an assignment of each input point to one of the $k$ centers while minimizing some cost, which is a function of the distances between points and centers. Different cost functions define different clustering objectives to be minimized. This paper focuses on the popular $k$-center clustering problem ($k$-center problem, for short), which aims at minimizing the maximum distance between a point and its assigned center.

A very natural assignment strategy for $k$-center associates each point with its closest center [12]. Imagine, however, that each point is a representation of some features of individuals, and that clustering implies decisions that may impact individual livelihoods. In this scenario, the decisions being made, i.e., the clustering, should not have a *disproportionate* effect on the people involved. For instance, people from a particular protected group cannot be segregated in a single cluster. This intuition is captured by the notion of *disparate*

impact [11]: people in different protected classes should not experience disproportionately different outcomes. Blindly ignoring protected attributes, however, is no solution [10]: correlated features (e.g., height which correlates with biological sex) can leak information about the protected attributes and may influence the clustering, leading to *unfair* solutions. This suggests that to achieve fairness in the clustering we need to explicitly take into account protected attributes when assigning points to centers.

The study of fair $k$-clustering under the disparate impact notion has been initiated by Chierichetti et al. [9] and generalized in subsequent works [3, 5, 19]. Each point is assigned one or more colors to model the protected attributes, and the clustering has to be built so that in each cluster the fraction of points of each color is within a color-specific range. For instance, if the input set has half blue points and half red points, each cluster could be required to have roughly half blue points and half red points. State of the art approaches to fair clustering with multiple colors are based on Linear Programming, which limits their scalability to large datasets. Coresets are an effective way of dealing with scalability issues for big data analytics [15]. A coreset is a compact representation of a large instance on which computationally demanding (e.g., LP-besed) algorithms can be run to efficiently obtain good solutions for the whole instance. For fair clustering in the big data setting, coresets have been recently used in [4] to reduce the size of the linear programs, yielding a 2-pass streaming algorithm and a 2-round MapReduce/MPC algorithm, attaining, respectively, $(7 + \varepsilon)$ and 9 approximations.

### 1.1 Our contribution

In this paper, we present an improved coreset-based strategy for fair $k$-center clustering of multi-colored points in general metrics, whose accuracy/performance tradeoffs are analyzed in terms of the doubling dimension of the data set. We devise implementations of our strategy in the sequential, streaming and MapReduce/MPC frameworks, yielding the following contributions, where $S$, $\Gamma$, and $D$ represent, respectively, the input dataset, the set of colors, and the doubling dimension of $S$.

- A sequential algorithm for fair $k$-center which attains a $(3 + \varepsilon)$ approximation, and whose running time is linear in $|S|$ for constant $k$, $|\Gamma|$, $\varepsilon$, and $D$. (See Theorem 4.5 for the general statement.)
- A 2-pass streaming algorithm for fair $k$-center which attains a $(3 + \varepsilon)$ approximation and requires working memory which, for constant $k$, $|\Gamma|$, $\varepsilon$, and $D$, is $O(\log(d_{max}/d_{min}))$, where $d_{min}$ and $d_{max}$ are, respectively, the minimum and maximum pairwise distance in $S$. (See Theorem 5.1 for the general statement.)
- A 5-round MapReduce/MPC algorithm for fair $k$-center which attains a $(3 + \varepsilon)$ approximation and requires a local memory which, for constant $k$, $|\Gamma|$, $\varepsilon$, and $D$, is $O(\max\{|S|/p, p\})$, when $p$ processors are used. (See Theorem 6.1 for the general statement.)

*Conference'17, July 2017, Washington, DC, USA*
© 2023 Association for Computing Machinery.
ACM ISBN 978-x-xxxx-xxxx-x/YY/MM. . . $15.00
https://doi.org/10.1145/nnnnnnn.nnnnnnn

As in [3, 4], all of the above algorithms return solutions where the color distribution in each cluster complies with the fairness constraints within a modest additive violation of $4\Delta + 3$, where $\Delta \leq |\Gamma|$ is the maximum number of colors per point.

We implemented and ran our algorithms on real-world datasets, scaling up to 16 million points, to assess the effectiveness and scalability of our coreset-based strategy. The experiments show that our algorithms return solutions whose quality is comparable to the best attained by state-of-the-art algorithms but exhibit significantly better performance.

The main novelty of our approach is that it adapts (obliviously) to the dimensionality of the input dataset, becoming extremely accurate (abating considerably the approximation ratios of [4]), and time and space efficient for low-dimensional datasets, in all computational settings. Also, our experiments provide evidence of its practicality.

**Structure of the paper.** The rest of the paper is organized as follows. Section 2 summarizes the relevant related work. Section 3 formally defines the problem and states some basic technical facts. Sections 4, 5, and 6 describe and analyze, respectively, our sequential, streaming and MapReduce algorithms. Finally, our experimental results are reported in Section 7. For space limitations, some technical details are reported in an appendix.

## 2 RELATED WORK

For space limitations, in this section we limit our literature review to the fair $k$-center clustering problem, in which a fair assignment to $k$ cluster centers has to be built while minimizing the maximum distance of a point from its assigned center. For a survey of other fair clustering objective functions and notions, we refer the interested reader to the recent tutorial[1] offered at *AAAI 2022*.

In the pioneering work by Chierichetti et al. [9], each point of the input is colored either red or blue, and a feasible solution is an assignment that preserves the balance of colors in each cluster, i.e., the ratio of blue to red points in each cluster must be the same as the ratio in the input dataset. The authors provide a combinatorial algorithm yielding a 3 approximation for the $k$-center objective. The main limitation of their approach, however, is that it is limited to the case of a single binary protected attribute. An extension to the case where the protected attribute can take one out of many colors has been devised by Rösner and Schmidt [19], who provide a 14-approximation algorithm.

The notion of balance has been generalized by Bercea et al. in [5]. In their work, the ratio of each color in each cluster is allowed to take values within user-specified color-specific ranges. The paper proposes approaches based on Linear Programming, obtaining a 5 approximation for fair $k$-center with exact preservation of the ratios, and a bicriteria 3 approximation that incurs a small violation of the fairness constraints. The main drawback of their approach is that it generates linear programs with a number of variables *quadratic* in the size of the input set.

Bera et al. [3] provide a further extension to the fairness notion, allowing each point to have multiple colors (thus supporting the notion of multiple protected attributes). As in [5], the balance of each individual color in a cluster is then required to be within

pre-specified color-specific ranges. The paper proposes a two-step approach valid for all $k$-clustering problems with a $\mathcal{L}_p$ norm objective (thus including $k$-center, $k$-median, and $k$-means, among the others), where the centers are first identified using an *unfair* approximation algorithm for the unconstrained $k$-clustering objective and then the assignment of points to centers is obtained using an LP-based technique. For $k$-center, their approach leads to a 3 approximation[2], with an additive $4\Delta + 3$ violation of the fairness constraints, where $\Delta$ is the maximum number of colors of a point. Importantly, this approach reduces the number of variables in the LP program to $O(k \cdot n)$, where $n$ is the input size.

Coreset-based streaming and MapReduce/MPC instantiations of the aforementioned strategy are presented in [4]. In both cases, the approach still relies on first determining a good set of unfair centers, together with the determination of a weighted summary of the input set upon which a variant of the LP introduced in [3] is solved to identify a suitable assignment of points to centers. The resulting algorithms achieve a 2-pass $7 + \varepsilon$ approximation in the Streaming setting, and a 2-round 9 approximation for the MPC.

Ahmadian et al. [1] study a different $k$-center variant, where there is an upper bound $\alpha$ to the ratio of each color in each cluster, but there are no lower constraints on the ratios. They devise a 3-approximate LP-based solution to the problem, and they also provide a combinatorial 12-approximation algorithm for the special case of $\alpha = 0.5$.

The goal of reducing the size of the LP used to build the fair assignment of points is further pursued by Harb and Lam [14], that are thus able to achieve the same approximation factors as in [4] while being considerably faster in practice.

## 3 PRELIMINARIES

This section formally defines the problems studied in this paper, and states some important technical facts. Consider a metric space $(M, d)$. We will analyze the performance of our algorithms in terms of the dimensionality of the input set $S \subseteq M$ which, for general metric spaces, can be captured by the notion of *doubling dimension*, reviewed below.

For any $p \in S$ and $r > 0$, let the *ball of radius $r$ centered at $p$*, denoted as $B(p, r) \subseteq S$, be the subset of all points of $S$ at distance at most $r$ from $p$. Then, the *doubling dimension* of $S$ is the minimum value $D$ such that, for all $p \in S$, any ball $B(p, r)$ is contained in the union of at most $2^D$ balls of radius $r/2$ centered at points of $S$. The notion of doubling dimension has been used extensively for a variety of applications (e.g., see [7, 8, 13, 17] and references therein).

Given an input set $S$, we assume that each point $x \in S$ is colored with a *color combination* of at most $\Delta$ colors out of a set of colors $\Gamma$.[3] With $S_\ell \subseteq S$ we denote the set of points whose color combination contains $\ell \in \Gamma$. For $x \in S$ we use $col(x) \subseteq \Gamma$ to denote its color combination, and define $C_S \subseteq 2^\Gamma$ to be the family of all color combinations associated with at least one point in $S$.

A $k$-clustering of a set $S$ is a pair $(C, \phi)$ where $C \subseteq S$ is the set of *centers*, and $\phi : S \rightarrow C$ is the *assignment function* that maps each

---

[1]https://www.fairclustering.com/

[2]In fact, in [3] a 4 approximation is claimed, but a careful reading of the proof reveals that, for the $k$-center objective, the approximation factor can be brought down to 3.
[3]This allows to model the setting of multiple sensitive attributes.

point of $S$ to a center. The $k$-center cost, also called *radius*, of a $k$-clustering $(C, \phi)$ is the largest distance between a point and its assigned center:

$$r_{C,\phi}(S) = \max_{x \in S} d(x, \phi(x))$$

Given a set of centers, the standard, color-oblivious way of building a clustering is by assigning each point to its closest center (with ties broken arbitrarily). Let this assignment function be denoted by $\phi_{unf}(\cdot)$, where *unf* stands for *unfair*, and let $OPT_{unf}(S, k)$ be the minimum radius of any $k$-clustering of $S$ under $\phi_{unf}(\cdot)$. (We will omit $S$ and $k$ when clear from context.)

For the unfair $k$-center problem, the classic $O(kn)$-time algorithm by Gonzalez [12] provides a 2 approximation. The algorithm, which we refer to as GMM (Greedy Minimum Maximum) implements the following simple greedy strategy. The set of centers is initialized with an arbitrary point. Then, the next center is selected to be a point at maximum distance from all previously selected centers. The procedure is repeated until there are $k$ centers.

Our analysis will make use of the following result, which was proved in [6, Lemma 1].

LEMMA 3.1. *Let $X \subseteq S$. For a given $k$, let $T_X$ be set of $k$ centers computed by GMM on $X$. We have*

$$r_{T_X, \phi_{unf}}(X) \leq 2 \cdot OPT_{unf}(S, k)$$

Clearly, the aforementioned standard assignment function might lead to unfair results, in the sense that different clusters might exhibit different proportions of points with the same color. This motivates the following additional constraint. A clustering $(C, \phi)$ for $S$ is called *fair* if, for each $\ell \in \Gamma$, for given parameters $\beta_\ell \leq \alpha_\ell$:

$$\beta_\ell \leq \frac{|\{x \in S_\ell : \phi(x) = c_i\}|}{|\{x \in S : \phi(x) = c_i\}|} \leq \alpha_\ell \quad \forall c_i \in C.$$

In other words, fairness requires that the fraction of points whose color combination includes color $\ell$ is between parameters $\beta_\ell$ and $\alpha_\ell$ in every cluster. Our algorithms will enforce this notion of fairness within some (small) tolerance. More precisely, as in [3] we say that a clustering $(C, \phi)$ for $S$ is *fair with additive violation* $\lambda$ if for every $c_i \in C$ and $\ell \in \Gamma$,

$$\beta_\ell \cdot |\{x \in S : \phi(x) = c_i\}| - \lambda \leq |\{x \in S_\ell : \phi(x) = c_i\}|$$

and

$$|\{x \in S_\ell : \phi(x) = c_i\}| \leq \alpha_\ell \cdot |\{x \in S : \phi(x) = c_i\}| + \lambda.$$

Note that the fairness conditions are stated for each color *independently*. This means that a point with multiple colors will be involved in multiple fairness constraints. An alternative approach would be that of considering every color combination in $C_S$ as a new, different color and enforcing a fairness constraint for each of these new colors. Clearly, this simpler approach can be modeled as the case of a single color per point. It is important to observe that no fair clustering may exist for a given multicolored pointset $S$ under a certain set of fariness constraints.

An optimal fair clustering is a fair clustering which minimizes the radius, denoted as $OPT_{fair}(S, k)$. (In case no fair clustering exists, we set $OPT_{fair}(S, k) = +\infty$.) The following basic fact trivially holds.

FACT 1. *For a given set $S$ and any fairness constraint, we have*

$$OPT_{unf}(S, k) \leq OPT_{fair}(S, k)$$

## 3.1 Big-data models of computation

In the MapReduce model [18], an algorithm executes in a sequence of *parallel* rounds. In each round a multiset $X$ of *key-value* pairs is transformed in a new multiset $Y$ by means of a *mapper* function, followed by the application of a *reducer* function to obtain a final multiset $Z$. Crucially, the local memory available to each *mapper* and *reducer* is limited by a parameter $M_L$, whereas the *aggregate* memory across all mappers and reducers is limited by parameter $M_A$. An algorithm in this model strives to minimize the number of rounds while complying with the memory limits.

We emphasize that our MapReduce algorithms admit a straight-forward porting to the MPC model [2], maintaining the same round and space complexity. Hence, all the results in this paper stated for MapReduce hold identically for the MPC model.

## 4 SEQUENTIAL ALGORITHM

This section describes our sequential coreset-based algorithm for fair k-center clustering. The input consists of a set of colored points $S$, the number of clusters $k$, the fairness constraints, represented by the two vectors $\alpha = \{\alpha_\ell : \ell \in \Gamma\}$ and $\beta = \{\beta_\ell : \ell \in \Gamma\}$, and an accuracy parameter $\varepsilon \in (0, 1)$. The algorithm executes three main steps. In the first step, a small coreset $T$ of colored and weighted points is computed from $S$, so that each $x \in S$ has a *proxy* $\pi(x) \in T$ with the same color combination, and each $t \in T$ carries a weight denoting the number of original points for which it acts as a proxy. In the second step, a solution $C \subseteq T$ consisting of $k$ centers is computed running GMM on $T$, and a skeleton of the final clustering is computed by suitably distributing the weights of the coreset points among the centers. Finally, in the third step, the skeleton is turned into the final clustering. The pseudocode for this high-level structure of the algorithm is depicted in Algorithm 1. The three steps and the procedures that they use are described in detail in the following subsections.

---

**Algorithm 1:** Sequential Fair k-Center Clustering

**Input:** Set of points $S$, parameters $k$, $\alpha$, $\beta$ and $\varepsilon$
**Output:** Set of centers $C$, assignment function $\phi$

```
/* Step 1                                          */
(T, π) ← CORESETCONSTRUCTION(S, k, ε);
/* Step 2                                          */
C ← GMM(T, k);
φ̂ ← WEIGHTDISTRIBUTION(T, C, α, β);
/* Step 3                                          */
φ ← FINALASSIGNMENT(φ̂, S, T, C);
return (C, φ);
```

---

## 4.1 Step 1: Coreset construction

We build the weighted coreset $T$ as follows (see Algorithm 2 for the pseudocode). First we run $k$ iterations of GMM on $S$ to determine a set of $k$ centers, which we denote as $T^k$, and compute the radius $r_{T^k, \phi_{unf}}$. Then, we continue to run GMM until the first iteration $\tau \geq k$ such that

$$r_{T^\tau, \phi_{unf}} \leq (\varepsilon/6) \cdot r_{T^k, \phi_{unf}}.$$

**Algorithm 2: CORESETCONSTRUCTION**

**Input:** Set of points $S$, parameters $k$ and $\varepsilon$

/* Identify coreset points                         */

$T \leftarrow \{$an arbitrary point of $S\}$;

**while** $|T| < k$ **do** $T \leftarrow T \cup \{\arg\max_{x \in S} d(x, T)\}$ ;

$r_k \leftarrow \max_{x \in S} d(x, T)$;

**while** $\max_{x \in S} d(x, T) > (\varepsilon/6)r_k$ **do**

$\quad \lfloor\ T \leftarrow T \cup \{\arg\max_{x \in S} d(x, T)\}$;

/* Build the proxy function and weights          */

**for** $t \in T$ **do**

$\quad$ Build $|C_S|$ copies of $t$ with distinct color combinations;

$\quad$ Set the weight $w(t)$ of each copy to $0$;

**for** $x \in S$ **do**

$\quad t' \leftarrow \arg\min_{t \in T:col(t)=col(x)} d(x, t)$ ;

$\quad w(t') \leftarrow w(t') + 1$;

$\quad \pi(x) \leftarrow t'$;

**return** $T, w, \pi$;

---

Now, for each point $t \in T^{\tau}$ we create $|C_S|$ copies, each colored with a distinct color combination in $C_S$. The resulting set of $|C_S| \cdot |T^{\tau}|$ copies will be our coreset $T$. Then, we determine a proxy function $\pi : S \to T$ which assigns to each point $x \in S$ the closest coreset point of the same color combination, namely

$$\pi(x) = \arg\min_{t \in T : col(t)=col(x)} d(x, t) \qquad \forall x \in S.$$

Also, for each coreset point $t \in T$ we compute a weight $w(t)$, corresponding to the number of points of $S$ for which $t$ is a proxy :

$$w(t) = |\{x \in S : \pi(x) = t\}| .$$

Points of $T$ with zero weight (i.e. which are the proxy of no input point) are simply discarded. Observe that all the points proxied by the same coreset point $t$ have the same color combination.

The following lemma upper bounds the distance between each input point from its representative in $T$.

LEMMA 4.1. *Let $T$ be the coreset constructed above for the set $S$, and let $\pi$ be the associated proxy function. Then, for each $x \in S$ we have:*

$$d(x, \pi(x)) \le (\varepsilon/3) \cdot OPT_{unf}$$

PROOF. We have that

$$r_{T^{\tau},\phi_{unf}} \le (\varepsilon/6) \cdot r_{T^k,\phi_{unf}}$$
$$\le (\varepsilon/3) \cdot OPT_{unf}$$

where the first inequality holds by construction, and the second by Lemma 3.1. □

We now bound the size of the coreset.

LEMMA 4.2. *If $S$ has doubling dimension $D$, then*

$$|T| \le |C_S| \cdot k \cdot (12/\varepsilon)^D$$

PROOF. We first prove an upper bound on the number $\tau$ of iterations needed by GMM to obtain a radius

$$r_{T^{\tau},\phi_{unf}} \le (\varepsilon/6) \cdot r_{T^k,\phi_{unf}}.$$

Consider the unfair $k$-center clustering induced by $T^k$ using $\phi_{unf}$, whose radius is $r_{T^k,\phi_{unf}}$. By the doubling dimension property, each of the $k$ clusters can be covered using at most $(12/\varepsilon)^D$ balls of radius $\le (\varepsilon/12)r_{T^k,\phi_{unf}}$, for a total of at most $h = k \cdot (12/\varepsilon)^D$ balls.

Consider now the execution of $h$ iterations of GMM on $S$, with $T^h$ being the set of centers and $x \in S$ being the point farthest from any center in $T^h$. It is easy to see that GMM ensures that any two points in $T^h \cup \{x\}$ are at distance at least $r_{T^h,\phi_{unf}}$ from one another. Since two of these points must fall into one of the $h$ balls mentioned above, by the triangle inequality we have that

$$r_{T^h,\phi_{unf}} \le 2(\varepsilon/12) \cdot r_{T^k,\phi_{unf}} = (\varepsilon/6) \cdot r_{T^k,\phi_{unf}}$$

Hence, we are guaranteed that after running $h$ iterations of GMM we find a set of points meeting the stopping condition, which implies $\tau \le h$. The lemma follows by noting that each point in $T^h$ is replicated at most $|C_S|$ times in $T$. □

## 4.2 Step 2: creating the clustering skeleton

Recall that coreset $T$ computed in Step 1 is such that each $t \in T$ represents $w(t)$ points of $S$ with the same color combination, which, by virtue of Lemma 4.1, are rather close to $t$. In Step 2, our algorithm first computes a set $C$ of $k$ centers by running GMM on $T$, and then invokes a procedure called WEIGHTDISTRIBUTION, described below, to distribute the weight of each $t \in T$ among one or more centers of $C$, so to minimize the maximum distance between coreset points and one of the centers receiving their weights (which we will refer to as the *radius of the distribution*) while, at the same time, enforcing the fairness constraints. This distribution will be modeled through a weight assignment function $\hat{\phi} : T \times C \to \mathbb{N}$ which will provide a skeleton of the final clustering and will be used in Step 3 to extract the assignment function $\phi$.

To achieve the aforementioned weight distribution, we make use of a weighted version of the Frequency Distributor LP of Harb and Shan [14]. Let $R$ be a guess on the radius of the distribution and consider the power set $2^C$ of the set of centers $C$. For each color combination $L \in C_S$ and each subset of centers $C' \in 2^C$, define $J_{C',L,R}$ as the set of points with color combination $L$ that are within distance $R$ from all and only the points of $C'$, namely

$$\begin{aligned} J_{C',L,R} \;=\; & \{t \in T : col(t) = L \wedge d(t, c) \le R \; \forall c \in C' \wedge \\ & \wedge \; d(t, \overline{c}) > R \; \forall \overline{c} \notin C'\} \end{aligned}$$

Each $J_{C',L,R}$ is referred to as a *joiner* in [14]. For a joiner $J$, we introduce the following notation: $C_J$ denotes the subset of centers defining $J$, $col(J)$ denotes the color combination common to all of its points and

$$w(J) = \sum_{t \in J} w(t)$$

denotes the total weight carried by the points of the joiner. Let $\mathcal{J}(R)$ be the set of joiners obtained for the guess $R$ and observe that they define a partition of $T$. For every $\ell \in \Gamma$ we also define

$$\mathcal{J}(R)_{\ell} = \{J \in \mathcal{J}(R) : \ell \in col(J)\}$$

The crucial observation is that a joiner $J \in \mathcal{J}(R)$ acts as a *super point* in the sense that the weight of any of its points can be indifferently distributed to *any* center in $J_C$. Thus, the weight

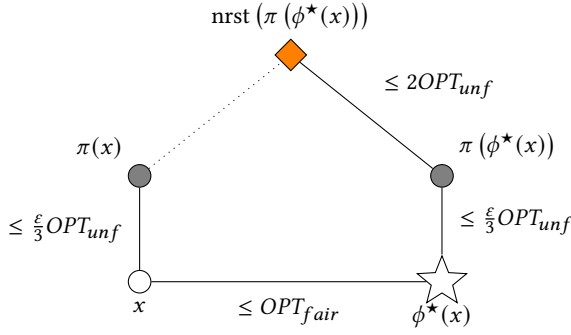

**Figure 1**

distribution can be computed at the joiner level rather than the coreset point level. To this purpose, the following linear program is defined which uses a variable $z_{J,c}$ for every joiner $J$ and center $c \in C_J$.

**LP-WFD**$(\mathcal{J}(R), C)$

$$z_{J,c} \geq 0 \qquad\qquad J \in \mathcal{J}(R), c \in C_J \qquad (1)$$

$$\sum_{c \in C_J} z_{J,c} = w(J) \qquad\qquad \forall J \in \mathcal{J}(R) \qquad (2)$$

$$\beta_\ell \sum_{J \in \mathcal{J}(R)} z_{J,c} \leq \sum_{J' \in \mathcal{J}(R)_\ell} z_{J',c} \qquad \forall c \in C, \ell \in \Gamma \qquad (3)$$

$$\sum_{J' \in \mathcal{J}(R)_\ell} z_{J',c} \leq \alpha_\ell \sum_{J \in \mathcal{J}(R)} z_{J,c} \qquad \forall c \in C, \ell \in \Gamma \qquad (4)$$

Condition (2) ensures that all the weight is assigned to some center, whereas Conditions (3) and (4) encode the fairness constraints. Note that the fairness constraints (3) and (4) are defined cluster-wise, and can be specified through variables $z_{J,c}$ which exploit the aggregation of the points defined by the joiners.

By construction, in a feasible solution to the above LP the nonzero $z_{J,c}$'s define an association between joiners and centers, such that for every coreset point $t$ belonging to some joiner $J$ and any center $c$ with $z_{J,c} > 0$, we have $d(t, c) \leq R$. Clearly, for small values of $R$ no feasible solution may exist. The following lemma provides a crucial lower bound to values $R$ which yield feasible solutions.

LEMMA 4.3. *Suppose that the weighted coreset $T$ computed for $S$ features a proxy function $\pi$ such that $d(x, \pi(x)) \leq \varepsilon/3$, for every $x \in S$. Then, for $R \geq (3 + (2/3)\varepsilon)OPT_{fair}$, the linear program above has a feasible solution.*

PROOF. Consider the set of centers $C$ selected by GMM, and the optimal fair clustering $(C^\star, \phi^\star)$ of cost $OPT_{fair}$. We will show that we can distribute the weight of points in $T$ to points in $C$ within distance $R$ so that the constraints of the linear program are satisfied.

For each point $x \in S$, consider its optimal fair center $\phi^\star(x)$, and let nrst$(\pi(\phi^\star(x)))$ be the center of $C$ nearest to $\pi(\phi^\star(x))$. By Lemma 3.1, we have that $d(\text{nrst}(\pi(\phi^\star(x))), \pi(\phi^\star(x)) \leq 2OPT_{unf}$. Figure 1 depicts all the points involved, along with relevant bounds on their distances. Let $J \in \mathcal{J}$ be the joiner such that $\pi(x) \in J$. By following the chain of inequalities of Figure 1, we have that $d(\pi(x), \text{nrst}(\pi(\phi^\star(x)))) \leq R$, hence

nrst$(\pi(\phi^\star(x))) \in C_J$. We determine values for the variables of **LP-WFD**$(\mathcal{J}(R), C)$ by "moving" one unit of weight from $\pi(x)$ to the variable $z_{J, \text{nrst}(\pi(\phi^\star(x)))}$. After processing all points in $S$, it is immediate to see that the group of constraints (2) is satisfied.

As for the fairness constraints (3) and (4), for $c \in C^\star$, let $C^\star(c) = \{x \in S : \phi^\star(x) = c\}$ be the optimal cluster centered in $c$. Similarly, let $C_\ell^\star(c) = \{x \in S_\ell : \phi^\star(x) = c\}$ for $c \in C^\star$, $\ell \in \Gamma$ be the set of points of color $\ell$ assigned to the cluster centered in $c$. Clearly, each optimal cluster $C^\star(c)$ must respect the fairness constraints, i.e.

$$\beta_\ell \leq \frac{|C_\ell^\star(c)|}{|C^\star(c)|} \leq \alpha_\ell \qquad (5)$$

for each $c \in C^\star$ and $\ell \in \Gamma$. Now, for each $c \in C$ let $N(c) = \{c^\star \in C^\star : \text{nrst}(\pi(c^\star)) = c\}$ be the set of optimal centers for which $c$ is the closest center in $C$ to their proxy in the coreset. By the weight assignment procedure described above, we have that any center $c \in C$ is assigned a weight equal to the number of points in $\cup_{c' \in N(c)} C^\star(c')$. Therefore we have that for any color $\ell \in \Gamma$ and any center $c \in C$

$$\beta_\ell \leq \frac{\sum_{c' \in N(c)} |C_\ell^\star(c')|}{\sum_{c' \in N(c)} |C^\star(c')|} \leq \alpha_\ell$$

by Fact 2 (in the Appendix) and Inequality (5), which proves that the set of constraints (3) and (4) are also satisfied. □

In Step 2, after computing the centers $C$ we run Procedure WEIGHTDISTRIBUTION which performs the following operations (see Algorithm 3 for the pseudocode). First it computes and sorts the $|T|k$ distances between the coreset points and the centers, and then performs a binary search over these distances to identify the smallest value $R$ such that the LP-WFD yields a feasible solution $Z_{LP-WFD} = \{z_{J,c} : J \in \mathcal{J}(R) \wedge c \in C\}$. Note that this solution may be fractional. In order to derive an integral weight assignment to the centers, we run Procedure CORESETASSIGN. The procedure first transforms $Z_{LP-WFD}$ into an integral solution $Z_{LP-WFD}^{int} = \{z_{J,c}^{int} : J \in \mathcal{J}(R) \wedge c \in C\}$, by using the iterative rounding procedure presented in [4], and then derives the weight assignment function $\hat{\phi}$ by distributing the weight of the coreset points of each joiner $J$ among the centers, as specified by the $z_{J,c}^{int}$'s. The rounding introduces a mere additive violation of the fairness constraints, as stated in the following lemma. For space limitations, the details of Procedure CORESETASSIGN and the proof of the lemma are moved to Appendix B.

LEMMA 4.4. *Procedure CORESETASSIGN returns a weight distribution function $\hat{\phi} : T \times C \to \mathbb{N}$ such that for every color $\ell \in \Gamma$ and every center $c \in C$*

$$\beta_\ell \cdot \sum_{t \in T} \hat{\phi}(t, c) - (4\Delta + 3) \leq \sum_{t \in T_\ell} \hat{\phi}(t, c) \leq \alpha_\ell \cdot \sum_{t \in T} \hat{\phi}(t, c) + (4\Delta + 3),$$

*where $T_\ell$ is the subset of coreset points whose color combination contains $\ell$.*

### 4.3 Step 3: Final assignment

In the last step, the algorithm uses the weight assignment function $\hat{\phi}$ computed in Step 2 to compute the final assignment function $\phi$

**Algorithm 3:** WEIGHTDISTRIBUTION

**Input:** Weighted coreset $T$, set of centers $C$, parameters $\alpha, \beta$

**Output:** Assignment $\hat{\phi} : T \times C \to \mathbb{N}$

$\Lambda \leftarrow$ Sorted list of distances $d(t, c)$, $\forall t \in T, c \in C$;

Do binary search on $\Lambda$ to find the smallest $R$ such that

    LP-WFD$(\mathcal{J}(R), C)$ yields a feasible solution

    $z^* = \{z_{J,c} : J \in \mathcal{J}(R), c \in C_J\}$;

$\hat{\phi} \leftarrow$ CORESETASSIGN$(T, \mathcal{J}(R), C, z^*)$;

---

**Algorithm 4:** FINALASSIGNMENT

**Input:** Sets $S, T$ and $C$, and assignment $\hat{\phi} : T \times C \to \mathbb{N}$

**Output:** Assignment $\phi : S \to C$

**for** $x \in S$ **do**

    $t \leftarrow \pi(x)$;

    $c \leftarrow$ arbitrary $c \in C : \hat{\phi}(t, c) > 0$;

    $\phi(x) \leftarrow c$;

    $\hat{\phi}(t, c) \leftarrow \hat{\phi}(t, c) - 1$;

**return** $\phi$;

---

between the original points of $S$ and the centers in $C$. Observe that, by construction, $\hat{\phi}$ ensures that for each $t \in T$

$$\sum_{c \in C} \hat{\phi}(t, c) = w(t) = |\{x \in S : \pi(x) = t\}|.$$

Therefore, we can compute $\phi$ through a sequential scan of $S$, where for each $x \in S$ an arbitrary center $c$ with $\hat{\phi}(\pi(x), c) > 0$ is chosen, and $\phi(x)$ is set equal to $c$ while $\hat{\phi}(\pi(x), c)$ is decreased by 1. The pseudocode is depicted in Algorithm 4.

### 4.4 Putting all pieces together

The following theorem concludes the analysis.

**THEOREM 4.5.** *For an input set $S$ of doubling dimension $D$, the above sequential algorithm returns a $(3 + \varepsilon)$-approximation to the optimum fair $k$-center clustering, with an additive violation $\leq 4\Delta + 3$ of the fairness constraints. For fixed values of $k$, $D$, and $|\Gamma|$, the algorithm requires linear time in the input set size.*

**PROOF.** The compliance with the fairness constraints is an immediate consequence of Lemma 4.4 and the derivation of $\phi$ from $\hat{\phi}$. As for the radius, the above assignment procedure, combined with the result of Lemma 4.3, ensures that for every $x \in S$

$$d(x, \phi(x)) \leq d(x, \pi(x)) + \max_{c \in C, \bar{z}_{\pi(x), c} > 0} d(\pi(x), c)$$

$$\leq (\varepsilon/3) OPT_{unf} + (3 + (2\varepsilon/3)) OPT_{fair}$$

$$\leq (3 + \varepsilon) OPT_{fair}.$$

The running time of the algorithm is dominated by the run of GMM to identify $T$, the cost of solving $O(\log(|T|k))$ instances of **LP-WFD**$(\mathcal{J}(R), C)$, and the cost of computing the final assignment. By Lemma 4.2 we have that $|T| \leq |C_S| \cdot k \cdot (12/\varepsilon)^D$, and by adapting the analysis in [14], we have that each **LP-WFD**$(\mathcal{J}(R), C)$ entails $O\left(k \cdot \min\{2^k|C_S|, |T|\}\right)$ variables and

$O\left(k\left(|\Gamma| + \min\{2^k|C_S|, |T|\}\right)\right)$ constraints. Thus, since $|C_S| \leq 2^{|\Gamma|}$, for fixed values of $k$, $D$, and $|\Gamma|$, the algorithm exhibits only linear dependence in $|S|$. □

It is important to remark that our algorithm attains an approximation factor that can be made arbitrarily close to the one of [3] but, for wide ranges of the involved parameters, reduces dramatically the size of the linear programs required to compute the solution, which dominate by far the computation costs.

## 5 STREAMING ALGORITHM

In this section, we describe a 2-pass streaming implementation of the sequential algorithm (Algorithm 1). We now regard the input $S$ as a stream of points. The first pass constructs the weighted coreset $T$ and, at the end of the pass, Step 2 of Algorithm 1, whose space requirements are independent of the stream size, is performed as is, returning the weight distribution function $\hat{\phi}$. Then, in the second pass the final assignment $\phi$ is computed.

The coreset construction requires the knowledge of the smallest and largest pairwise distances in the stream (or suitable approximations), denoted respectively as $d_{min}$ and $d_{max}$[4]. The first pass runs in parallel several *instances* for geometric guesses $R$ of the optimal radius of $OPT_{unf}(S, k)$, namely $R = 2^j d_{min}$, with $0 \leq j \leq \lceil \log_2(d_{max}/d_{min}) \rceil$. Let $S_i$ be the set of the first $i$ points of $S$. For $i \geq 1$, each instance maintains two sets of points:

- A set $C_R$ of up to $k + 1$ points with $d(x, C_R) \leq 2R$, $\forall x \in S_i$,
- A set $T_R$ of weighted points with $d(x, T_R) \leq \frac{\varepsilon}{12}R$, $\forall x \in S_i$.

$C_R$ is used to detect when the guess $R$ is too small, while $T_R$ is the candidate coreset. For each point $x$ in the stream, if $d(x, C_R) > 2R$, then $x$ is added to $C_R$. In case the size of $C_R$ exceeds $k$, this instance fails because the guess $R$ is too small. Otherwise, $x$ is processed as follows. If $d(x, T_R) > \frac{\varepsilon}{12}R$, then we add $x$ to $T_R$ with weight 1, and also add $|C_S| - 1$ copies of $x$ to $T_R$, with the other color combinations from $C_S$ and weights 0. If instead $d(x, T) \leq \frac{\varepsilon}{12}R$, then we take the point $t \in T_R$ which arrived the earliest (rather than the closest), and such that $col(t) = col(x)$ and $d(x, t) \leq \frac{\varepsilon}{12}R$, and increase $w(t)$ by one, thus making $t$ proxy of $x$. It is important to remark that we do not store the proxy function explicitly, since it would require linear memory. By using the earliest valid coreset point as the proxy, the proxy function can be reconstructed on the fly, a fact that will be used in the second pass of the algorithm. We select the output $\langle C_R, T_R, R \rangle$ of the non-failing instance associated with the smallest guess $R$. The pseudocode for the first pass is depicted as Algorithm 5

As mentioned above, at the end of the first pass, Step 2 of Algorithm 1 is run on $T$ to compute the weight distribution function $\hat{\phi}$ based on $(C_R, T_R)$. In the second pass, the final assignment $\phi$ is computed using the naturally streamlined algorithm FINALASSIGN-MENT (Algorithm 4) with the only difference that, for every $x \in S$, its proxy $\pi(x)$ is obtained as the earliest coreset point $t$ such that $col(t) = col(x)$ and $d(x, t) \leq (\varepsilon/12)R$. We have:

**THEOREM 5.1.** *For an input stream $S$ of doubling dimension $D$, the above 2-pass algorithm returns a $(3 + \varepsilon)$ approximation to the*

---

[4]A similar assumption was needed in the streaming algorithm by [4]. The assumption can be removed by introducing an extra pass (details will be provided in the full version).

**Algorithm 5:** Streaming coreset construction

**Input:** Stream of points $S$, parameters $k$ and $\varepsilon$
**Output:** Weighted coreset $T$ and radius $R$
**for** $R \leftarrow d_{min}, 2 \cdot d_{min}, 4 \cdot d_{min}, \ldots$ **do** in parallel
$\quad$ $C_R, T_R \leftarrow \emptyset$;
$\quad$ **for** $x \in S$ **do**
$\quad\quad$ **if** $d(x, C_R) > 2R$ **then** $C_R \leftarrow C_R \cup \{x\}$ ;
$\quad\quad$ **if** $|C_R| > k$ **then** **Fail** ;
$\quad\quad$ **if** $d(x, T_R) \leq \frac{\varepsilon}{12} R$ **then**
$\quad\quad\quad$ $t \leftarrow earliest(\{t \in T_R : d(x, t) \leq \frac{\varepsilon}{12} R \wedge col(t) =$
$\quad\quad\quad$ $col(x)\})$;
$\quad\quad\quad$ $w(c) \leftarrow w(c) + 1$;
$\quad\quad$ **else**
$\quad\quad\quad$ Add a copy of $x$ to $T_R$ for each color
$\quad\quad\quad$ combination;
$\quad\quad\quad$ Set $w(t)$ to 1 for the copy with the same color
$\quad\quad\quad$ combination as $x$, and to 0 for all others;

**return** $\langle C_R, T_R, R \rangle$ where $R$ is the min non-failed guess;

optimum fair k-center clustering, with an additive violation $\leq 4\Delta + 3$ of the fairness constraints using working memory

$$O\left(k \cdot |C_S| \left((24/\varepsilon)^D \log(d_{\max}/d_{min}) + |\Gamma| \min\{2^k, k \cdot (24/\varepsilon)^D\}\right)\right)$$

PROOF. First, we prove that the set $C_R$ returned by Algorithm 5 provides a 4 approximation to the unfair $k$-center problem. Consider the smallest integer $j$ such that $d_{min} \cdot 2^{j-1} \leq r_k^* \leq d_{min} \cdot 2^j$, where $r_k^*$ is the radius of an optimal solution to unfair $k$-center on the stream $S$, and define $\hat{R} = 2^j$. Clearly, $\hat{R} \leq 2r_k^*$ Observe that the instance associated to guess $\hat{R}$ indeed terminates successfully, since the points put in $C_{\hat{R}}$ must necessarily belong to different optimal unfair clusters. Also, at the end of the stream, we will have that for each $x \in S$, $d(x, C_{\hat{R}}) \leq 2\hat{R} \leq 4 \cdot r_k^*$. Therefore, Algorithm 5 will return a triple $\langle C_R, T_R, R \rangle$ with $R \leq \hat{R} \leq 4 \cdot r_k^*$. Consider now coreset $T_R$. For each $x \in S$, we have

$$d(x, T_R) \leq (\varepsilon/12)R \leq (\varepsilon/12)4r_k^* \leq (\varepsilon/3)OPT_{unf},$$

hence $T_R$ has the same quality of the coreset computed by the sequential algorithm, and the approximation guarantee exhibited by the final solution can thus be argued similarly.

Let us now bound the working memory required by the streaming algorithm. By virtue of the doubling dimension property, for every instance associated with a generic guess $R$ and until the instance is non-failed, each of the $\leq k$ clusters of radius $2R$ induced by $C_R$ can be covered by using at most $(24/\varepsilon)^D$ clusters of radius $\varepsilon R/12$, and each such cluster may contribute $|C_S|$ coreset points to $T_R$. Also, in the first pass, we have $\log(d_{\max}/d_{min})$ instances of the algorithm running in parallel. The bound on the working memory is a consequence of the bounds on the $|T_R|$'s and on the size of the linear programs executed at the end of the first pass. $\square$

## 6 MAPREDUCE ALGORITHM

In this section, we adapt the sequential strategy presented in the Section 4 to the distributed setting, devising the following 5-round

**Table 1: Datasets used in the experimental evaluation.**

| dataset | n | d | $|\Gamma|$ | dataset | n | d | $|\Gamma|$ |
|---|---|---|---|---|---|---|---|
| hmda | 16 007 906 | 8 | 18 | adult | 32 561 | 5 | 7 |
| census1990 | 2 458 285 | 66 | 8 | creditcard | 30 000 | 14 | 7 |
| athlete | 206 165 | 3 | 2 | bank | 4 521 | 9 | 3 |
| diabetes | 89 782 | 9 | 5 | victorian | 4 500 | 10 | 45 |
| 4area | 35 385 | 8 | 4 | reuter_50_50 | 2 500 | 10 | 50 |

MapReduce algorithm. In the first round, the input $S$ is partitioned arbitrarily across the $p$ workers, and worker $i$ extracts a subset $T_i$ of points by executing the first two while-loops of Algorithm 2 on its partition, using accuracy parameter $\varepsilon/2$ rather than $\varepsilon$. In the second round, the $T_i$'s are gathered in a single worker, and their union, say $T'$, is further processed through the first two while-loops of Algorithm 2, using again accuracy $\varepsilon/2$, to extract a subset $T'' \subset T'$. In the third round, a copy of $T''$ is sent to each worker, which makes $|C_S|$ copies of each $t \in T''$ and computes their weights with respect to the points of $S$ in its partition, as specified in the last two for-loops of Algorithm 2. In the fourth round, the final coreset $T$ is built by gathering all copies of the points of $T''$ created by the different workers in a single worker, and coalescing the $p$ like-colored copies of each $t \in T''$ by adding up their weights. This produces the final weighted coreset $T$, on which Step 2 of Algorithm 1 is run sequentially to compute the set $C$ of centers and $p$ projections $\hat{\phi}_i$, $i \in [p]$, of the weight distibution function $\hat{\phi}$, relative to the $p$ partitions of $S$. The final assignment is then built in the fifth round, by sending to the $i$-th worker the projection $\hat{\phi}_i$, so that procedure FINALASSIGNMENT can be applied independently within its partition. The following theorem, whose proof is deferred to Appendix C for lack of space, summarizes the accuracy-space tradeoffs featured by the above algorithm.

THEOREM 6.1. For an input set $S$ of doubling dimension $D$, the above 5-round MapReduce algorithm returns a $(3 + \varepsilon)$ approximation to the optimum fair k-center clustering, with an additive violation $\leq 4\Delta + 3$ of the fairness constraints, using local memory

$$M_L = O\left(\max\left\{\frac{|S|}{p}, k|C_S|\left(p\,(24/\varepsilon)^D + |\Gamma|\min\left\{2^k, k\,(24/\varepsilon)^D\right\}\right)\right\}\right)$$

and linear aggregate memory.

## 7 EXPERIMENTS

Our experiments aim at: **(a)** comparing the performance of different algorithms for different values of $k$ in terms of radius and running time; **(b)** verifying the influence of the coreset size on the quality of the approximation; **(c)** demonstrating the efficiency of the streaming and MapReduce approaches on large datasets. We compare our approach against the following baselines: UNFAIR, the classic GMM algorithm [12], which returns the unfair clustering radius that we use as a reference point; BERA-ET-AL, the algorithm from [3]; KFC, the algorithm of [14]; BERA-ET-AL-STREAM and BERA-ET-AL-MR the streaming and MapReduce algorithms from [4], respectively. We devised best-effort implementations of all of the above algorithms, always improving on the running time of the original ones while maintaining the same accuracy, but for KFC, for which we used the author's code. We experiment with the same datasets used by


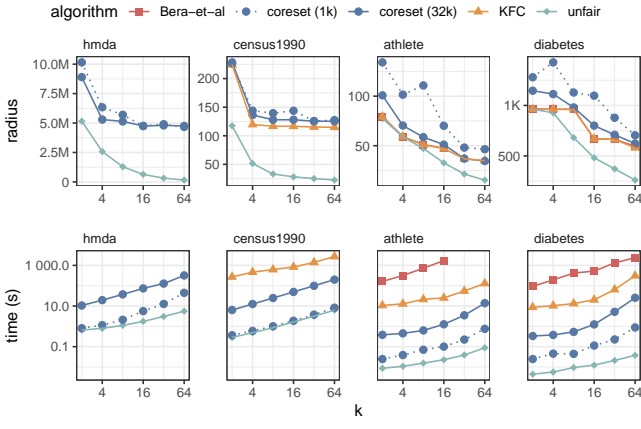

**Figure 2: Radius (top) and running time (bottom, in log scale) of different algorithms vs. $k$ (in logarithmic scale). Missing points are for timed-out runs.**

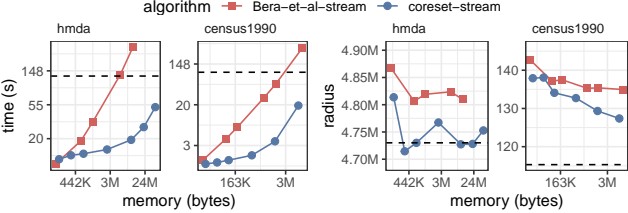

**Figure 3: Streaming algorithms performance vs. memory (log scale): time (left, log scale) and radius (right).**

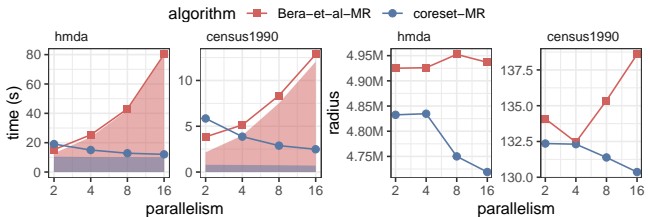

**Figure 4: MapReduce algorithms performance vs. parallelism: time (left) and radius (right).**

previous works [3, 4, 14], whose features are given in Table 1. Due to space constraints, in this section we report only on the four largest ones, providing the results for the others in Appendix D, along with details about our experimental setup. As in [3], we set strict fairness constraints: $\beta_\ell = r_\ell(1 - \delta)$ and $\alpha_\ell = r_\ell/(1 - \delta)$, for $r_\ell = |S_\ell|/|S|$ and $\delta = 0.01$. For our algorithms, rather than governing the coreset size indirectly through $\varepsilon$, we fix it directly as a multiple of $k$, allowing for more interpretable results. Our source code is publicly available (see https://anonymous.4open.science/r/fair-clustering-C8EF/.)

*Sequential setting.* Figure 2 reports the radius and the running time of different sequential algorithms for $k = 2^i$, $i \in [1, 6]$. UNFAIR always has the smallest radius and the fastest running time, as expected. The best fair clustering radius is up to 15 times larger (hmda) than the unfair clustering radius. Notably, while an unfair clustering sees its radius constantly decreasing with $k$, for some datasets (hmda, census1990) the fair radius tends to remain constant as $k$ becomes larger. We observed that in these cases the fairness constraints encourage the assignment of the majority of points to a few ($\ll k$) large clusters, whose radius remains large irrespective of the value of $k$. Two instances of our algorithm, dubbed CORESET, were run with coreset sizes $k$ and $32k$. As expected, using a larger coreset gives a clustering with a smaller radius, which becomes comparable (at most 1.39 times larger) to the one attained by KFC and BERA-ET-AL. Noticeably, the slight increase in the radius is compensated by the significantly faster execution time (Figure 2, bottom), even with a coreset of size $32k$. Indeed, in our experiments, BERA-ET-AL timed out after one hour on census1990 and hmda, and on athlete for large $k$, whereas KFC timed out on hmda, with 16 million points. In contrast, our coreset-based algorithm completed just in under 5 minutes for $k = 64$ and a coreset of size $32k$.

*Streaming.* We compare our algorithm (CORESET-STREAM) with BERA-ET-AL-STREAM [4] for different amounts of memory allowed to both algorithms, and for $k = 32$. For CORESET-STREAM, larger memory implies that each of the $\log_2 d_{max}/d_{min}$ instances of the algorithm builds a larger coreset, whereas for BERA-ET-AL-STREAM,

larger memory implies that a smaller $\varepsilon$ is used, hence more parallel instances are run, each building a $k$-clustering. Both implementations feature the same level of optimization. Figure 3 reports the running time and the radius achieved by both algorithms on the two largest datasets of the testbed. The dashed lines, used for reference, mark the best running time and radius attainable by the sequential fair algorithms. We observe that for comparable memory usage, our CORESET-STREAM algorithm runs faster than BERA-ET-AL-STREAM. As for the radius, CORESET-STREAM provides a radius closer to the best radius found by sequential algorithms. The figure highligths the fundamental tradeoff of our coreset construction: larger coresets allow for better approximations. Interestingly, for small memories, both algorithms are faster than the fastest sequential one. This is due both to the low aspect ratio $d_{max}/d_{min}$ ($\approx 56K$ for hmda, $\approx 43$ for census1990) and to the streaming clustering strategy which may require less than $n$ distance computations per center.

*MapReduce.* We compare our algorithm (CORESET-MR) with BERA-ET-AL-MR [4] for different numbers of processors and $k = 32$. Figure 4 reports the results in terms of time and radius. The line in the time plots marks the total running time, whereas the shaded area represents the time required to solve the linear program on the pointset created by each algorithm. As already noted in [4], the running time of BERA-ET-AL-MR increases with the number of processors because it is dominated by the time to solve the linear program (shaded red area) whose size increases with the number of processors, thus annulling scalability. Conversely, in CORESET-MR the size of the linear program is independent of the number of processors: in fact, the blue shaded area marks a constant running time for all processor counts. Consequently CORESET-MR features good scalability. As for the radius, both approaches provide solutions of comparable quality.

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

# A TECHNICAL FACT

FACT 2. *Let $a_1, \ldots, a_k$ and $b_1, \ldots, b_k$ be two sequences of numbers. Then*

$$\min_i \frac{a_i}{b_i} \leq \frac{\sum_i a_i}{\sum_i b_i} \leq \max_i \frac{a_i}{b_i}$$

PROOF. Let $M = \max_i \frac{a_i}{b_i}$. By definition we have $a_i/b_i \leq M$, i.e. $a_i \leq M b_i$ for any $i$. Therefore $\sum_i a_i \leq M \sum_i b_i$, which implies

$$\frac{\sum_i a_i}{\sum_i b_i} \leq M = \max_i \frac{a_i}{b_i}.$$

By similar reasoning one can prove the lower bound. □

# B PROCEDURE CORESETASSIGN

The output of the linear program **LP-WFD** is fractional, and thus needs to be rounded. To this end, we use the iterative rounding procedure presented in [4], which in turn is based on the approach proposed in [3] and which we summarize here for completeness.

Let $Z_{LP-WFD} = \{z_{J,c} : J \in \mathcal{J}(R) \lor c \in C\}$ be a feasible (fractional) solution of the **LP-WFD** problem. Following [4] we define the following new variables:

$$\overline{w}(J) = w(J) - \sum_{c \in C_J} \lfloor z_{J,c} \rfloor \qquad \forall J \in \mathcal{J} \qquad (6)$$

$$Z_c = \sum_{J \in \mathcal{J}} \left( z_{J,c} - \lfloor z_{J,c} \rfloor \right) \qquad \forall c \in C \qquad (7)$$

$$Z_{c,\ell} = \sum_{J' \in \mathcal{J}_\ell} \left( z^\star_{J',c} - \lfloor z_{J',c} \rfloor \right) \qquad \forall \ell \in \Gamma, c \in C \qquad (8)$$

Variables $\overline{w}(J)$ are the residual weight of $J$ to distribute after rounding (note that they are integer, as $w(J)$ is integer).

Using these variables, we can set up the following linear program. **LP-RES:**

$$0 \leq \bar{z}_{J,c} \leq 1 \qquad \forall J \in \mathcal{J}, \forall c \in C_J \qquad (9)$$

$$\sum_{c \in C_J} \bar{z}_{J,c} = \overline{w}(J) \qquad \forall J \in \mathcal{J} \qquad (10)$$

$$\lfloor Z_c \rfloor \leq \sum_{J \in \mathcal{J}} \bar{z}_{J,c} \leq \lceil Z_c \rceil \qquad \forall c \in C \qquad (11)$$

$$\lfloor Z_{c,\ell} \rfloor \leq \sum_{J' \in J_\ell} \bar{z}_{J',c} \leq \lceil Z_{c,\ell} \rceil \qquad \forall c \in C, \forall \ell \in \Gamma \qquad (12)$$

The solution of the above linear program might still be fractional. There are two key insights. First, for some $J, c$ the variable $\bar{z}_{J,c}$ could be assigned 0 or 1. If the variable is 1, then we can increase $\hat{z}_{J,c}$ by one. In either case, the variable $\bar{z}_{J,c}$ can be removed from the linear program. Second, if the residual weight for a $J \in \mathcal{J}(R)$ is small, then removing from the linear program the fairness constraints involving $J$ leads only to a small fairness violation.

Given the above observations, the linear program **LP-RES** is solved iteratively until there are no more variables. Bera et al. show how the resulting integral solution $Z^{int}_{LP-WFD} = \{z^{int}_{J,c} : J \in \mathcal{J}(R) \lor c \in C\}$ incurs an additive violation of the fairness constraints of up to $4\Delta + 3$ [4, Theorem 3.6]. In particular, we have that $\forall \ell \in \Gamma$ and $\forall c \in C$

$$\beta_\ell \sum_{J \in \mathcal{J}} z^{int}_{J,c} - 4\Delta - 3 \leq \sum_{J' \in \mathcal{J}_\ell} z^{int}_{J',c} \leq \alpha_\ell \sum_{J \in \mathcal{J}} z^{int}_{J,c} + 4\Delta + 3 \qquad (13)$$

---

**Algorithm 6: CORESETASSIGN**

/* Round the solution                                    */
$z_{J,c}^{int} \leftarrow \lfloor z_{J,c} \rfloor \; \forall J \in \mathcal{J}$;
Define $\overline{w}$, $Z_c$, and $Z_{c,\ell}$ as in equations (6), (7), and (8);
Construct LP-RES;
**while** $\exists J \in \mathcal{J}(R) : \sum_{c \in C} \hat{z}_{J,c} \neq \overline{w}(J)$ **do**
    $\overline{z} \leftarrow$ solution to LP-RES;
    **foreach** $\overline{z}_{J,c} = 0$ **do**
        Remove $\overline{z}_{J,c}$ from LP-RES
    **foreach** $\overline{z}_{J,c} = 1$ **do**
        Remove $\overline{z}_{J,c}$ from LP-RES;
        $z_{J,c}^{int} \leftarrow \hat{z}_{J,c} + 1$;
        Decrease by 1 $Z_c$ and $Z_{c,\ell}$;
    **foreach** $c \in C$ **do**
        **if** $|\{J \in \mathcal{J} : 0 < \overline{z}_{J,c} < 1\}| \leq 3$ **then**
            Remove from constraints (11) involving $c$;
    **foreach** $c \in C, \ell \in \Gamma$ **do**
        **if** $|\{J' \in \mathcal{J}_\ell : 0 < \overline{z}_{J',c} < 1\}| \leq 3$ **then**
            Remove from RES constraints (12) involving $c$;

/* Build the weight distribution                         */
**foreach** $J \in \mathcal{J}, c \in C_J$ **do**
    **foreach** $t \in J$ **do**
        $\hat{\phi}(t,c) \leftarrow \min\{w(t), z_{J,c}^{int}\}$;
        $z_{J,c}^{int} \leftarrow z_{J,c}^{int} - \hat{\phi}(t,c)$;
**return** $\hat{\phi}(\cdot, \cdot)$;

---

Algorithm 6 shows the implementation of this iterative rounding procedure. The last step of Algorithm 6 builds the weight assignment function. By construction we have that for each center, the total weight assigned to $c \in C$ by means of the function $\hat{\phi}$ is

$$\sum_{t \in T} \hat{\phi}(t,c) = \sum_{J \in \mathcal{J}} z_{J,c}^{int} \qquad \forall c \in C$$

and similarly, for any given color $\ell \in \Gamma$

$$\sum_{t' \in T_\ell} \hat{\phi}(t',c) = \sum_{J' \in \mathcal{J}_\ell} z_{J',c}^{int} \qquad \forall c \in C, \forall \ell \in \Gamma$$

Therefore, we have that Inequality (13) holds for the weight distribution function as well:

$$\beta_\ell \sum_{t \in T} \hat{\phi}(t,c) - 4\Delta - 3 \leq \sum_{t' \in T_\ell} \hat{\phi}(t',c) \leq \alpha_\ell \sum_{t \in T} \hat{\phi}(t,c) + 4\Delta + 3$$

and thus Lemma 4.4 follows.

## C   PROOF OF THEOREM 6.1

For $i \in [p]$, let $S_i \subseteq S$ be the subset of the input set $S$ assigned to the $i$-th worker, and let $\pi_i : S_i \to T_i$ be the proxy function associated to the coreset $T_i \subseteq S_i$ extracted locally at each worker. Letting $T_i^k \subseteq T_i$ be the first $k$ centers computed by GMM on $S_i$, by Lemma 3.1 we have that $r_{T_i^k, \phi_{unf}} \leq 2OPT_{unf}$, whence $r_{T_i, \phi_{unf}} \leq$

$((\varepsilon/2)/6) \cdot 2OPT_{unf} \leq (\varepsilon/6) \cdot OPT_{unf}$. Recall that $T' = \cup_{1 \leq i \leq p} T_i$. It follows that for each $x \in S$:

$$d(x, T') \leq (\varepsilon/6)OPT_{unf}.$$

The argument can now be repeated identically with respect to the extraction of the coreset $T''$ from $T'$. Thus, we have that for each $t \in T'$, $d(t, T'') \leq (\varepsilon/6) \cdot OPT_{unf}$. Consider now the final coreset $T$ computed in the fourth round, and, for each $x \in S$ let $\pi'(x)$ be the point in $T'$ closest to $x$. We have that

$$d(x, T) \leq d(x, \pi'(x)) + d(\pi'(x), T) \leq (\varepsilon/3)OPT_{unf}.$$

Observe that coreset $T$ satisfies the hypotheses of Lemma 4.3, which in combination with Theorem 4.5 ensures the approximation factor.

For what concerns the bound on the local space, the local space requirements per round is as follows: $O(|S|/p)$ for Round 1, $O\left(kp(24/\varepsilon)^D\right)$ for Round 2, $\max\{|S|/p, k(24/\varepsilon)^D|C_S|\}$ for Round 3, $\max\{k|C_S|p(24/\varepsilon)^D, k|\Gamma||C_S|\min\{2^k, k(24/\varepsilon)^D\}\}$ for Round 4, and $\max\{|S|/p, k^2(24/\varepsilon)^D\}$ for Round 5. The bound on the local space follows by maximizing over the space requirements of each round.

## D   ADDITIONAL EXPERIMENTS

In this appendix we report the experiments and the information omitted from the main paper for space reasons.

### D.1   Experimental setup

We implement all the algorithms using Python 3.11.4, leveraging the implementations provided by Harb and Lam [14]. Performance-sensitive parts such as the coreset construction (in the sequential, streaming, and MapReduce settings) are implemented using Rust 1.70.0 for efficiency, and made available to the Python code via Python bindings. Rust provides a similar level of control to C++, and thus allows to write code that is much more efficient than the Python equivalent [16].

We use CPLEX 22.1.1.0 to solve the linear programs.

The streaming and MapReduce implementations of the algorithms of [4] do not appear to be publicly available. We therefore re-implemented them using Rust, applying the same optimizations as on our own code.

All the code used to carry out the experimental evaluation is publicly available[5]. Furthermore, the code repository provides information to download and preprocess all the datasets.

### D.2   Experiments

Figure 5 reports the results for the experiment described in paragraph *Sequential setting* of Section 7 for all the datasets reported in Table 1. Panels in the Figure are arranged by decreasing size of the corresponding dataset.

The same takeaways discussed in the main paper apply: increasing the coreset size improves the quality of the radius of the clustering found by CORESET; the running time is faster than the baselines; the solution quality of CORESET is comparable with the baselines. We note that on smaller datasets the gap with the baselines in terms of running time is less marked. This is not surprising, since for small datasets the linear programs used by the baselines are

---

[5]https://anonymous.4open.science/r/fair-clustering-C8EF/

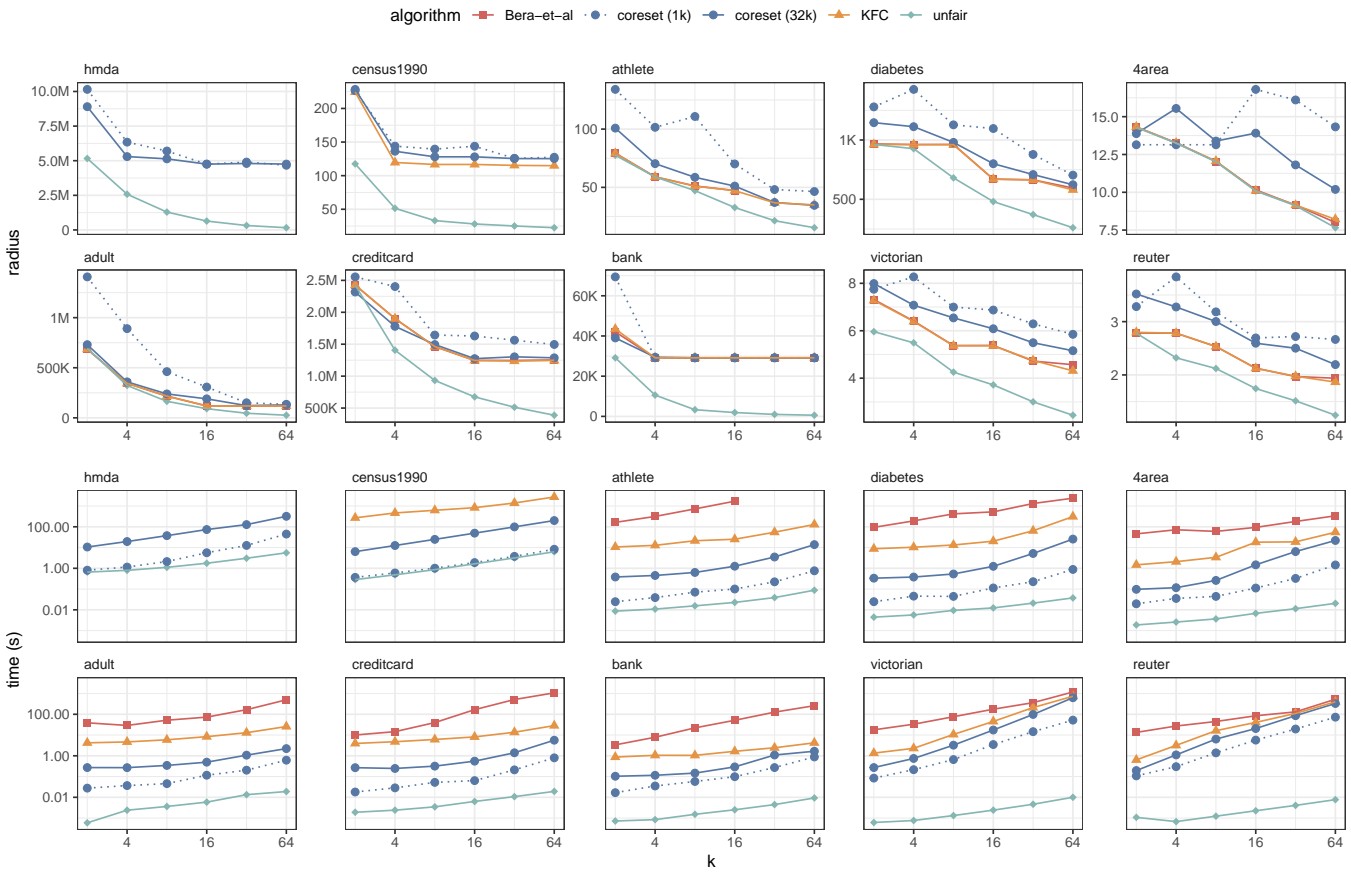

**Figure 5: Performance of sequential algorithms in terms of radius (top two rows of plots) and running time (bottom two rows of plots, in logarithmic scale) against *k* (in logarithmic scale)**

small enough to allow a fast solution. We stress, however, that our coreset construction scales to large instances.

As for the streaming and MapReduce settings, we remark that in Section 7 we report results on the two largest datasets in the testbed. Given that the other datasets are comparatively very small, they do not provide any meaningful insight on the behavior of the MapReduce and Streaming algorithms.

