# OpenReview forum: "Fast and Accurate Fair $k$-Center Clustering in Doubling Metrics"
_ACM.org/TheWebConf/2024/Conference — TheWebConf24 Oral_

### Official Review · Reviewer_dGeV · 2023-11-18

**Novelty:** 4
**Technical Quality:** 5

**Review:**

The authors study the problem of designing efficient streaming/parallel algorithms for fair k-center. There is a substantial	literature on fair k-center where one wants to minimize the max distance of a point to the cluster center subject to cluster constraints on the distributions of color classes of points in each cluster. The authors provide new a algorithm that achieves almost the best approx. in sequential setting (3+eps) while being efficient in streams and parallel setting for a regime where points have low dimensions, low number of color combinations. The algorithms are based on standard results in coreset theory for bounded doubling dimension spaces and on prior work on fair clustering (including using LP). The results are shown to be practical in experiments for a set of standard datasets used in the literature that don’t have high dimensions.

All in all, it is an interesting paper. On the negative side the algorithmic results are based on standard methods and the algorithm scales exponentially with number of colors, and dimension of the space.

--- rebuttal ---
I have read the reply of the authors and I confirm my rating of the paper.

**Questions:**

Lemma 4.3.  is this d(x, pi(x)) <= eps/3 * OPT_unfair instead of <= eps/3?

**Reviewer Confidence:**

3: The reviewer is confident but not certain that the evaluation is correct

**Scope:**

3: The work is somewhat relevant to the Web and to the track, and is of narrow interest to a sub-community

---

### Official Review · Reviewer_ysdz · 2023-11-23

**Novelty:** 6
**Technical Quality:** 7

**Review:**

This paper studies the fair k-center problem, providing a faster coreset based approach and improved approximation algorithms in the streaming and MapReduce settings.

Strengths:

* The coreset approach is a theoretical interesting new approach for the problem that is shown to be effective in practice
* The paper significantly improves the best approximation factors for streaming (from 7+eps to 3+eps) and MapReduce/MPC (9 approx in 2-round to 3+eps approx in 5 rounds)
* In addition to strong theoretical results, the experiments section is very well done and the authors have provided new fast implementations for methods (including faster more optimized implementations of previous methods and an implementation for a method whose implementation does not appear to be open source)
* Overall the paper is very well written

Weaknesses:

* This is minor, but it would be nice to include some more details about the level of fairness each of the methods satisfies. I'm a little bit confused about some aspects of this; see question below.

Minor typo I caught:

* LP-besed --> LP-based (page 1)

**Questions:**

I'm just a little bit confused about how well fairness constraints are actually satisfied by different methods. In the technical portion of the paper you mention "additive violation \lambda" and in the introduction you talk about an additive violation 4\Delta + 3. In other words, these methods (and previous ones) have small violations in the fairness constraints. This is perfectly acceptable, but then I would be curious to see in practice how much these violate the constraints in practice. In the experiments section though, you state "we set strict fairness constraints" and no results are shown for how much the methods violate constraints. Can you reconcile this for me? If in practice the constraints are satisfied even if they are not always in theory, it would be nice to have more details about this. If the methods do still violate constraints, then it would be helpful to additionally evaluate the methods in terms of how well they satisfy these fairness constraints, as this is another tradeoff to consider.

**Reviewer Confidence:**

3: The reviewer is confident but not certain that the evaluation is correct

**Scope:**

3: The work is somewhat relevant to the Web and to the track, and is of narrow interest to a sub-community

---

### Official Review · Reviewer_VaCw · 2023-11-24

**Novelty:** 4
**Technical Quality:** 5

**Review:**

Generally, it shows that loads of work has been done on the specific area and gives out satisfying performance. The explanation of the main algorithms is in clear detail, and most of the proofs directed to the final conclusion are convincing. In some of the scenarios, this work outscores the current benchmark by a visible margin.
The content of the paper is pretty solid, but the importance of the factors and methods led by this paper is not exhibited completely. The proofs carried out in the main part of the paper seems kind of redundant in descriptions, which to some degrees makes the organization of the paper imbalanced.
From the view of the applications of the work, we hold the view that this work takes a step forward. It shows that the method taken here is referable in the community, and of reasonable competitiveness. Yet the scope of the research is a bit narrow, and the innovative points are more like different paths to the results than advanced setups, and it remains unsolved whether the method is explainable for the improvement of the performance.

**Questions:**

From my side, there are still a few points in the paper that show somehow confusing. Overall, I don't really understand the relationship between your experiment results and the initial motivation that you introduced in Chapter 1. It doesn't mean that your experiments make no sense at all, but I personally think the organization of the part of introduction is a bit far away from the work itself, especially your novel contributions to the community. And the part for the experiments in the main paper is kind of short for an enhancement of the performance. Comparing to the algorithms, the part that shows the practical effect of the algorithms (which is stated of importance at the beginning) lacks instructions and reflection. It is recommended that you put some of the explanation to the design of algorithm into the Appendix, and let the readers clearly find out how the experiments are carried out and why to use the specific setup. Some more necessary analysis to the experiments and the results is also helpful to make this part more valuable.
For another info, the equation on ll. 447-448, Page 5 looks a bit weird. It seems like that some items are missing or unexpected. And Figure 1 in the paper should be clearer on both the picture and the footnote, especially when it is used in the proving session.
Looking forward to your upcoming examination and revision.

**Reviewer Confidence:**

1: The reviewer's evaluation is an educated guess

**Scope:**

2: The connection to the Web is incidental, e.g., use of Web data or API

---

### Official Review · Reviewer_3h6U · 2023-11-27

**Novelty:** 4
**Technical Quality:** 5

**Review:**

The submitted work studies a variant of the $k$-Center Clustering problem, where given a metric space and a set of points S, the standard version aims to find a set of k points in S (i.e. centers) and an allocation of points to centers, such that the radius of the clustering is minimized. The "fair" variant studied in this paper further assume each point comes with a set of colors (that is a subset of the ground set $\Gamma$), such that for each color $c$ in $\Gamma$, the fraction of points having color $c$ allocated to a center (relative to the total number of points allocated to that center) must fall between some lower and upper bounds.

Previous results can achieve a $3$-approximation while obeying the fairness constraints (up to small additive violation), and the submitted work gives algorithms with a slightly worse $(3+\epsilon)$-approximation but asymptotically faster running time when $k$, $|\Gamma|$, $\epsilon$ and the doubling dimension of the metric space are all small. The main idea is to construct a coreset by finding a large enough number of centers using the standard (unfair) algorithm and create a copy associated with each unique color (combination) of each center. Each point is then associated with the nearest point in the coreset with the same color combination. When $k$, $|\Gamma|$, $\epsilon$ and the doubling dimension of the metric space are all constants, the author(s) can prove that the required size of the coreset is also a constant, and thus one can run slow LP based algorithm on the coreset to find a fair clustering, and map it back to the original set of points.

The overall approach is fairly clean and well motivated, and the analysis is easy to follow. The empirical section does a reasonable job supporting the main claims of the paper.

**Questions:**

What does the $d$-column in Table 1 capture?

Do these datasets have qualitatively different doubling dimension. and do you observe the impact of that in experiment results?

In Figure 2, do you have the radius result for Bera-et-al in the cases when it finishes within time limit?

In your empirical evaluation, when you use a coreset of size 32K, what is the typical drop in radius in your coreset construction step, i.e. the ratio between the unfair radius of k iterations of GMM vs 32k iterations?

**Reviewer Confidence:**

2: The reviewer is willing to defend the evaluation, but it is likely that the reviewer did not understand parts of the paper

**Scope:**

3: The work is somewhat relevant to the Web and to the track, and is of narrow interest to a sub-community

---

### Decision · Program_Chairs · 2024-01-22

**Decision:**

Accept (Oral)

**Comment:**

The paper introduces a new algorithm for fair k-clustering and then discusses how to adapt it to obtain sequential, streaming and MapReduce implementations of their new approach. The paper complements the theoretical analysis with experimental results proving the effectiveness of their method.

 The paper is well-written and presents interesting contributions to a relevant topic for the community so it would be a nice addition to the conference program.